# The Complex Relationship between Hypoxia Signaling, Mitochondrial Dysfunction and Inflammation in Calcific Aortic Valve Disease: Insights from the Molecular Mechanisms to Therapeutic Approaches

**DOI:** 10.3390/ijms241311105

**Published:** 2023-07-05

**Authors:** Esmaa Bouhamida, Giampaolo Morciano, Gaia Pedriali, Daniela Ramaccini, Elena Tremoli, Carlotta Giorgi, Paolo Pinton, Simone Patergnani

**Affiliations:** 1Translational Research Center, Maria Cecilia Hospital GVM Care & Research, 48033 Cotignola, Italy; ebouhamida@gvmnet.it (E.B.); mrcgpl@unife.it (G.M.); gpedriali@gvmnet.it (G.P.); dramaccini@gvmnet.it (D.R.); etremoli@gvmnet.it (E.T.); 2Department of Medical Sciences, Laboratory for Technologies of Advanced Therapies (LTTA), University of Ferrara, 44121 Ferrara, Italy; carlotta.giorgi@unife.it

**Keywords:** calcific aortic valve stenosis, hypoxia, HIF-1α, mitochondria, oxidative stress, inflammation, therapeutic target

## Abstract

Calcific aortic valve stenosis (CAVS) is among the most common causes of cardiovascular mortality in an aging population worldwide. The pathomechanisms of CAVS are such a complex and multifactorial process that researchers are still making progress to understand its physiopathology as well as the complex players involved in CAVS pathogenesis. Currently, there is no successful and effective treatment to prevent or slow down the disease. Surgical and transcatheter valve replacement represents the only option available for treating CAVS. Insufficient oxygen availability (hypoxia) has a critical role in the pathogenesis of almost all CVDs. This process is orchestrated by the hallmark transcription factor, hypoxia-inducible factor 1 alpha subunit (HIF-1α), which plays a pivotal role in regulating various target hypoxic genes and metabolic adaptations. Recent studies have shown a great deal of interest in understanding the contribution of HIF-1α in the pathogenesis of CAVS. However, it is deeply intertwined with other major contributors, including sustained inflammation and mitochondrial impairments, which are attributed primarily to CAVS. The present review aims to cover the latest understanding of the complex interplay effect of hypoxia signaling pathways, mitochondrial dysfunction, and inflammation in CAVS. We propose further hypotheses and interconnections on the complexity of these impacts in a perspective of better understanding the pathophysiology. These interplays will be examined considering recent studies that shall help us better dissect the molecular mechanism to enable the design and development of potential future therapeutic approaches that can prevent or slow down CAVS processes.

## 1. Introduction

Heart valve diseases are a major contributor to cardiovascular morbidity and mortality worldwide. They affect more than 13% of the population aged over 75 years old and occur when any type of the four heart valves (tricuspid, pulmonic, mitral, and aortic valves) is damaged. Calcific aortic valve disease (CAVD) is defined as a slowly progressing condition that ranges from mild valve aortic sclerosis to severe calcifying aortic valve stenosis. This progression manifests in approximately 2% of individuals over 65 years old annually [1,2,3].

Calcific aortic valve stenosis (CAVS) remains one of the most rapidly increasing and common forms of heart valve disorders, prevailing in over 3.4% of the aged population, making it a major predominant and critical public health care and economic burden [4], and it entails 80% of the risk to develop heart failure (HF) or mortality by approximately 25% annually [5].

Despite the growing amount of evidence to better understand the factors underlying CAVS progression, today it remains unclear in terms of molecular and cellular mechanisms. Currently, no drug strategies exist to prevent or treat CAVS once symptoms occur and the only clinical option available is a surgical or transcatheter valve replacement. Since CAVS incidence increases with age, it becomes critically urgent to identify and emphasize the pathophysiological causes to ameliorate the current therapy. CAVS is a multifactorial and active process, with aging being the principal risk factor. Other relevant factors in CAVS have been reported including gender “male”, obesity, smoking, hypertension, increased triglyceride levels (TG), and high oxidative stress [6]. The aortic valve (AV) is largely an avascular tissue, where oxygen (O_2_) and nutrients within the AV occur via passive diffusion. With aging and the early phases of CAVD, the significant O_2_ demand of the inflammatory cells with the further thickening process of AV reduces and impedes O_2_ levels progressively, which turns the AV region hypoxic. Hypoxia-inducible factor 1 alpha subunit α (HIF-1α) is the primary sensor of hypoxia, and it mediates numerous responses and acts as a central modulator of several target genes in the human organism [7,8]. Studies have identified the expression of HIF-1α, a pro-angiogenic transcription factor, in the calcific leaflet nodule of aortic stenosis (AS) [9,10]. However, the further impact of HIF-1α at the onset of AV disorders is still to be explored. Hypoxia is reported as a promoter of angiogenesis. Indeed, it has been suggested that the involvement of HIF-1α induces the expression of vascular endothelial growth factor (VEGF), thereby stimulating neo-angiogenesis, a feature of valvular disorders, and increasing metabolic adaptation, affecting in return the valve phenotype [10,11]. Importantly, numerous pieces of evidence have documented the appearance of neovascularization throughout the progression of the calcification process [12,13]. The presence of neovessels exhibits an increased expression of vascular and intercellular adhesion molecules that have also been linked to the inflammatory response, bone development, and calcification progression [14,15].

Today, inflammation is still considered the main active player in the phases that precede calcification. It is deeply entwined among other major contributors such as mitochondrial dysfunctions, which have recently been identified as the main contributor to CAVS; however, cause–effect relationships are difficult to address.

This review aims to cover the most recent understanding of the impact of the hypoxia signaling pathways, mitochondrial dysfunction, and inflammation on a key condition of cardiovascular disease, which is “calcific aortic valve stenosis”. We will discuss the new molecular and cellular mechanisms involved in CAVS and will propose further critical hypotheses and interconnections on the complexity of these impacts with a perspective of better understanding the pathophysiology that shall help us to design further future potential therapeutic strategies in CAVS processes and research methodology. The following keywords were used to search the PubMed database: “inflammation and CAVS”, “mitochondria and CAVS”, and “Hypoxia signaling”. We also reviewed articles on the concepts of “inflammation and mitochondria” and “inflammation and hypoxia signaling”. Most recent original articles and review articles published and reported in the last four years were included.

## 2. Overview of the Calcific Aortic Valve Disease

### Epidemiology and Histological Structure

Calcific aortic valve disease (CAVD) is a progressive heart valve disorder defined by an active process of remodeling with an uncontrolled formation of calcium nodules, valve mineralization, which leads to the consequent narrowing of the valve, to the restriction of the valvular area, and to serious problems in the correct blood flow.

CAVD is an important clinical problem because it is the most common heart valve disease; its frequency increases with age, and men are more affected than women. According to epidemiological studies, 2.8% of adults over 75 years old have some degree of CAVS, and one adult over 65 out of four presented valvular sclerosis [16].

A very recent study focused on the epidemiology of CAVD during the past 30 years by taking full advantage of estimates from the Global Burden of Diseases, Injuries, and Risk Factors 2019 [17,18]. The results obtained confirmed that the global number of incident cases of CAVD has increased continuously over the past 30 years and the prevalence is higher in men than in women. CAVD mortality is associated with three primary causes: high systolic blood pressure, a diet high in sodium, lead exposure, and age are all fundamental factors in CAVD incidence [19].

To note, the most common congenital heart disease is bicuspid AV, affecting 0.5% to 1.4% of the population [20,21]. Most of these patients develop stenosis and valvular calcification early in their lifetime with a faster progression [22]; an event that might be linked to genetic, mechanical, and biological factors [23]. Given the previous data, the clinical approaches against CAVD have received enhanced interest over the last decade, but, right now, the gold standard intervention for CAVD patients is valve replacement or the percutaneous implantation of valve prostheses, such as mechanical prostheses and bioprostheses [24]. This approach is common, life-saving, and certainly improves the patient’s quality of life but might have some side effects, such as the necessity for long-term anticoagulation therapy or failure of bioprostheses due to tissue degeneration and mineralization [25]. The one-way blood flow from the left ventricle to the aorta goes through the aortic heart valve without regurgitation. The AV possesses three leaflets with a trilaminar conformation covered by endothelium: a layer named fibrosa, on the aorta side, a central layer named spongiosa, and a layer named ventricularis, on the left ventricle side [26] (Figure 1). Each layer is important for the mechanical feature of the AV and has a different tissue composition. Fibrosa is rich in collagen fibers indispensable for strength and support, the most present cell type is valvular interstitial cells (VICs). The spongiosa layer consists of glycosaminoglycans and is responsible for absorbing shear stress; it also presents smooth muscle cells (SMC) with contractile function. Ventricularis is made of collagen and elastin fibers, which provide the dynamism of the valve. The valve leaflets are attached to the annulus, a fibrous ring, fundamental for their connection to the aortic root and the dissipation of mechanical energy [27].

The main cellular components localized in the AV are VICs, present in all the layers, valvular endothelial cells (VECs) are on the surface of the leaflets, and there are a few SMCs, but only in the ventricularis layer [23]. VICs control matrix remodeling, they synthesize the extracellular matrix (ECM) and remodel collagen, maintaining normal valve structure; VECs are involved with VICs in the maintenance of the integrity of valve tissues [14,15].

CAVD is a multistep process that starts at the early stage with aortic sclerosis, characterized firstly by endothelial dysfunction, then by mechanical stress, lipid accumulation in the tissue and their subsequent oxidation, and then by initiating an inflammatory response with the infiltration of inflammatory cells, T cells, and macrophages [28]. A second phase is characterized by fibrosis and accelerated calcification; in fact, the inflammation process in stenotic valves leads VICs to differentiate myofibroblasts, which actuate matrix remodeling through the activation of matrix metalloproteinase (MMP) enzymes [29,30].

Along with stenosis, a key feature of this pathological condition is the calcification process, resulting from the activation of several osteogenesis pathways [31]. The increased lipid deposition induces VICs to change their phenotype through osteogenic differentiation, producing spheroid calcium phosphate particles, leading irreversibly to leaflet stiffening [32,33,34]. 

## 3. The Complex Interplay of Hypoxia Signaling, Mitochondrial Dysfunction, and Inflammation

### 3.1. Hypoxia Signaling and Molecular Regulation of HIF-1

Hypoxia plays a critical role in CVDs and is orchestrated by a hallmark heterodimer trans-acting DNA-binding hypoxia-inducible transcription factors (HIFs), which are key regulators mediating adaptation to hypoxic conditions and are modulated by an O_2_-sensitive-expressed alpha subunit (HIF-1α) (or its analogs HIF-2α and HIF-3α). In normal conditions, HIF-1α is continuously synthesized and hydroxylated through HIF prolyl-4-hydroxylases, leading to its rapid ubiquitination and proteasomal degradation (ubiquitin proteasome 26S), the von Hippel–Lindau (pVHL) function as a tumor suppressor binds to the ubiquitin ligase complex E3 targeting the HIF-1α subunit destruction in the O_2_ degradation domain, causing its short life. In contrast, under hypoxia, HIF-1α hydroxylation is suppressed through the inhibition of the O_2_-dependent propyl-hydroxylase-1, -2, and -3 enzyme activity (PHD1, -2, and -3), leading to the stabilization of HIF-1α in the cytosol, and migrates to the nucleus, where it forms a heterodimer with the beta subunits (HIF-1β, aryl hydrocarbon receptor nuclear translocator, ARNT) that bind to a core putative regulatory sequence called hypoxia response elements (HRE) with a consensus sequence (5′-RCGTG-3′) in the promoter or enhancer of target genes to enhance a concerted transcriptional response during a hypoxic condition [35] (Figure 2). Both α and β subunits have basic helix-loop-helix (bHLH) motifs, a DNA-binding domain that can bind HREs to target specific genes [8,36]. The transcription activity of the target genes requires not only the transfer of HIF-1α to the nucleus but also the complex HIF-1 requires the recruitment of multiple cofactors such as CREB-binding protein (CBP)/p300 and transcription intermediary factor 2 steroid-receptor activator that binds to the CTAD domain, and another cofactor that increases the HIF-1/HRE complex binding the M2 isoform of pyruvate kinase (PKM2) [37]. The canonical sensor of hypoxia, HIF-1α, mediates a cellular response during hypoxic conditions through the regulation of the transcription activity of enormous target genes, termed hypoxia-inducible genes encoding proteins, as examples: the lactate dehydrogenase-A (LDH-A) or pyruvate dehydrogenase kinase isoform 1 (PDK) [38,39]; VEGF-A [40]; erythropoietin (EPO) [41]; and inducible nitric oxide synthase (iNOS) [42], which are needed for improving tissue O_2_ homeostasis, energy metabolism, and efficient management of hypoxia-induced toxic stress, and elicit a crucial impact in various CVDs, such as ischemic heart disease (IHD) and HF [7,43]. HIF-2α, the analog of HIF-1α, is also termed endothelial PAS domain protein-1 (EPAS-1), and is predominantly enriched within endothelial cells and in highly vascularized tissues [44]. These two subunits (HIF-1α and HIF-2α) contain different spatial expressions in IHD, for example, and various effects in response to hypoxia [45]. Despite the recent interest in studying hypoxia and HIF-1 in CVDs, their role at the onset of CAVS remains unclear and requires deep further investigation.

### 3.2. Hypoxia Signaling and Inflammation in CAVD

The heart AV is an avascular tissue able to sustain metabolic activity, nutrition, and oxygenation through passive diffusion. Nevertheless, with age, the initiation of CAVS is exhibited by endothelial injury triggered by shear stress, lipid deposition, and inflammation [46]. In these early stages of CAVS, the thickening of the valve process compromises the diffusion of O_2_, resulting in tissue hypoxia, while the CAVS progression occurs through the abnormal remodeling of the ECM, which is modulated by the valve interstitial cells [47,48]. Sustained stimulation of hypoxia maintains HIF-1α signaling, leading to the upregulation of inflammation and fibrosis, but these effects are counterbalanced by sustained HIF-2α signaling, potentially linked to mitochondrial and peroxisomal abnormalities [49]. It is demonstrated that during hypoxia, HIF-1α and HIF-2α are activated and both can transactivate VEGF expression [50,51,52]. HIF-1α is reported to initiate the process of angiogenesis, while HIF-2α is required for vascular network maturation [53,54,55]. Nevertheless, it is not yet established whether HIF-2α contributes to the angiogenesis maturation and regenerative process in CAVS.

Hypoxia has been recently identified in both the aortic and mitral valves [56,57]. Moreover, diseased valvular interstitial cells in regions surrounding calcific nodules have been found to express HIF-1α, which is a pro-angiogenic transcription factor [9,10]. Additionally, in human aortic endothelial cells (HAVEC), the level of HIF-1α mRNA and protein expression are identified as elevated in response to disturbed flow as opposed to stable flow [58]. A study has demonstrated the significant upregulation of HIF-1α in the stenotic valves and its colocalization with angiogenesis in areas calcified [10]. These findings suggest a further key role of HIF-1α in influencing the valve phenotype through the induction of VEGF expression, thereby activating neo-angiogenesis, a feature of valvular disorders, to increase metabolic adaptation [10,59]. In contrast to the upregulation of pro-angiogenic factors, anti-angiogenic factors have been identified as suppressed [60,61,62]. Furthermore, there are five VEGF family growth factors that bind to specific tyrosine kinase receptors and play important roles in the formation of new blood vessels and lymphatic vessels [63,64]. Among the VEGF receptors is the soluble fms-like tyrosine kinase 1 (sFlt1), an anti-angiogenic component that sustains AV avascularity [65].

A recent study by Lewis and colleagues has shown that sFlt1 is expressed in the native normal AV, but its expression level is significantly downregulated in patients with CAVS [65]. The authors demonstrated for the first time the dual roles of hypoxia in stimulating angiogenesis in CAVS, as the classical way by inducing VEGF-A and by inhibiting the sFlt1 expression, which could elevate inflammation, thus contributing to CAVS progression [65]. Sphingosine 1-phosphate (SP1) is a bioactive lipid signaling mediator shown to inhibit angiogenesis via the activation of sFlt1 expression in CAVS patients [65]. However, the link between HIFs (HIF-1α and HIF-2α) and sFlt1 has not been studied yet in CAVS, and its mechanism of action in response to hypoxia is not clear. To our knowledge, the relationship between sFlt1 and HIFs in response to hypoxia has been studied only in the pathogenesis of Preeclampsia, the onset of hypertension during pregnancy [66,67]. The relationship between HIF-1 and sFlt1 needs to be studied to understand further the molecular mechanisms behind this process in CAVS. Prior studies have shown that sFlt1 induces inflammation in VICs when it synergizes with lipopolysaccharide (LPS), which is a major component of the outer membrane of Gram-negative bacteria [68]. LPS drives the activation of Toll-like receptor 4 (TLR4), a key receptor of innate and adaptive immunity, stimulating the inflammatory (interleukin 6 and 8 (IL-6, IL-8), and intercellular adhesion molecule-1 (ICAM-1)) and osteogenic responses (bone morphogenetic protein-2 (BMP-2) and runt-related transcription factor 2 (RUNX2) in CAVS patients [69,70]. IL-37 is a novel cytokine member of the IL-1 family and plays a potential role in suppressing inflammatory responses, identified as downregulated in CAVS [71,72]. The lower levels of IL-37 observed in CAVS patients have been explained by M1 macrophage infiltration in pathological AV stenosis (AVs). IL-37 inhibits the macrophages polarization M1 (reduction in IL-6, MCP-1, iNOS, and the surface marker of M1 (CD11c)) via the suppression of nuclear factor kappa B (NF-κB) and Notch homolog 1, translocation-associated (Notch1) signaling pathways [73]. Along with that, IL-37 is reported as a potent anti-osteogenic in AVICs from patients, through the suppression of the NF-κB and extracellular signal-regulated kinase (ERK1/2) [72]. The anti-inflammatory response of IL-37 is associated with NF-κB, suppressing the TLR4 ligand LPS-mediated IL-6, IL-8, monocyte chemoattractant protein-1 (MCP-1), and ICAM-1 stimulation in AVICs from patients [74], thus playing a critical role in CAVS physiopathology. Noteworthy, IL-37 is capable of inducing other anti-inflammatory pathways including the AMP-activated protein kinase (AMPK) and Phosphatase and tensin homolog, which may impact NF-κB stimulation in CAVS [71]. Morciano and colleagues have shown the correlation of both pro-inflammatory cytokine levels of IL-18 and IL-1β—which both belong to the IL-1 family—in CAVS [46]. Furthermore, IL-1β has been found to stimulate mTORC1, and, downstream, it enhances HIF-1α activity in several diseases [75]. The expression of the HIF-1α protein is induced by IL-1β in normoxia in multiple cell types [76,77,78]. For instance, HIF-2α is one among the different signaling pathways that are involved in M1 infiltration [79], but is yet to be studied in macrophages of AVICs. Previous studies have shed light on better understanding the further relationship between HIF-1, Signal Transducer and Activator of Transcription 3 (STAT3), and IL-37 in cancer disorders [80]. This would increase further interest in a deep understanding of the crosstalk of the IL-37 and HIF-1 in the pathogenesis of CAVS. Evidence has confirmed the upregulation of HIF-1α and its analog HIF-2α in valve stenosis. Interestingly, HIF-2α co-localizes with NF-κB in regions of calcified lesions of AS of patients [52]. These findings have been correlated positively with the enhanced levels of VEGF and the formation of neovessels [52]. On the other hand, it is well known that the expression of VEGF can also be increased by inflammatory cytokines [81]. The findings indicate further convergence between hypoxia and inflammatory mechanisms involved in the remodeling of the valve ECM, which contributes to VIC stimulation and calcification. It has been shown that the HIF-1α activates the expression of various proteins involved in ECM remodeling including the Neutrophil Gelatinase-Associated Lipocalin [82,83] and the MMP2 and MMP9, suggesting the key impact of hypoxia in ECM remodeling.

In light of the previous findings, a novel immune non-hypoxic process entailing the combination between LPS and interferon-γ (IFN-γ) has been explored to stimulate calcification in AVICs from patients, through the STAT1/HIF-1α signaling pathway [84]. Interestingly, this response to HIF-1α stimulation mediated by LPS is sex-dependent, as it is more robust in VICs from male donors compared to females [84]. The Janus kinases (JAK)-STAT signaling pathways are hallmark regulatory routes contributing to several cytokine responses activating mineralization and calcification of AV interstitial tissues, including IL-6 and IFN-γ [84,85,86]. Notably, the study of Parra-Izquierdo and colleagues also suggests further a relationship between JAK-STAT and HIF-1α-dependent sex differences in the context of CAVS [84]. However, females display lower responses and tend to be more protective compared with males due to the activation of the phosphatidylinositol 3-kinase (PI3K)-AKT signaling survival pathways [87]. Nevertheless, more research is needed to fully understand these relationships.

These findings propose the further potential involvement of HIF-1α in normoxia and in the early phases of CAVS when the hypoxic event is not yet activated; however, it is not known how LPS and/or IFN-γ could stabilize HIF-1α. Since the stabilization of HIF-1α is regulated by prolyl-4-hydroxylases under normoxic conditions, it is therefore possible that LPS and/or IFN-γ treatment could cause a decrease in hydroxylation, leading to the activation of HIF-1α. Other pioneer evidence supports the upregulation of HIF-1α in normoxic conditions—mediating the calcification process in AVICs. The ubiquitin E2 ligase C (UBE2C) is a member of the Anaphase Promoting Complex/Cyclosome (APC/C), which has been reported to also bind pVHL [88]. UBE2C upregulates the endothelial–mesenchymal transition (EndMT) and endothelial AV inflammation via the stimulation of HIF-1α levels through further ubiquitination and degradation of its upstream modulator pVHL, and this was accompanied by the reduction in microRNA-483–3p (miR-483) in HAECs [58]. In addition, the miR-483 mimics and the pharmacological suppressor of HIF-1α (PX478) significantly downregulate the porcine AV calcification through UBE2C reduction [58]. Other evidence supports the involvement of HIF-1α in valve calcification: PX478 significantly blocked the deposition of calcium resulting from distributed flow, and the response was more effective in male valve interstitial cells [10,11]. Other in vitro studies have identified the increased expression levels of HIF-1 signaling pathways including IL-6, HIF-1α, and Heme Oxygenase 1 (HMOX1) in CAVS-related ferroptosis signaling pathways [89]. Hence, one of the causes of ferroptosis is an iron overload that is involved in CAVS by enhancing calcium deposition and calcification in endothelial cells (HUVEC) [90]. It is identified that the endothelial cells play crucial impacts in the calcification process through the EndMT [91]. Multiple pathways are involved in this process including the signaling pathways involved in hypoxia and inflammation, such as the transforming growth factor beta (TGF-β) signaling pathway [92], and the Wnt signaling pathway [93,94].

In the same regard, inflammation is also triggered by fetuin-A (alpha2-Heremans Schmid glycoprotein); a 59 kDa glycoprotein synthesized in the liver, emerges as a potent circulating inhibitor of the calcification process, modulates macrophage polarization, and attenuates inflammation and fibrosis [95]. Intracellular fetuin-A suppresses calcification stimulated by transforming growth factor-β and bone morphogenetic proteins [96,97]. In addition, previous works have conflicting results on circulating fetuin-A as a biomarker for CAVD. Nevertheless, a meta-analysis demonstrated significantly lower levels of fetuin-A in AS patients compared to healthy conditions; also, fetuin-A levels have been significantly identified as being associated with CVD risk factors including age, male gender, smoking, low-density lipoprotein (LDL) and TG, hypertension, and diabetes [98]. Studies have shown an inverse correlation between fetuin-A levels and the progression of calcific AV and underlined diminished fetuin-A levels in AV sclerosis patients, the early asymptomatic phase of CAVD, suggesting that fetuin-A is an early calcification biomarker [99,100]. These studies suggest further involvement of fetuin-A in the initiation of AV calcification, hence raising the concept that fetuin-A, as an inhibitor of calcification, may prevent valvular calcifications when the calcium phosphate is disrupted, suggesting its correlation to calcium tissue deposition [101], and its levels, is also associated with inflammation and other comorbidities such as chronic kidney disease and diabetes [102]. In a very recent study, Chen et al. demonstrated the involvement of Fetuin-A in calcific osteogenic environment-induced VICs calcification, and in parallel, its level reported the decreased inhibition of miR-101 taking place [103]. However, the role of fetuin-A in the progression of CAVD has not been clearly investigated. Therefore, relationships of fetuin-A with tissue calcification and cardiovascular diseases in general are divergent, reflecting its diverse action. Notably, beyond the role of fetuin-A as a calcification suppressor in the serum phase, it acts as a potent calcium mineral scavenger, preventing the ectopic pathological calcification of the tissue especially in response to hypoxic stress in renal tissue remodeling upon IH injury [95]. Recently, fetuin-A has been identified as an evolutionary target gene of HIF-1 [104]; however, the correlation between HIF-1 and fetuin-A is still unclear in CAVS. Further understanding the complex interplay between HIF-1, fetuin-A, and pro-inflammatory cytokine in CAVS patients would pave the way to better predict the presence of CAVS and may provide further molecular targeting strategies.

### 3.3. Mitochondrial Impairment in CAVD

Mitochondria have key roles in eukaryotic cells, as they control several cellular mechanisms such as bioenergetics, signal transduction, and energy metabolism. Furthermore, these organelles are major regulators and executioners of cell death mechanisms, in particular autophagy and apoptosis [105]. To regulate these processes, cells have to preserve a functional mitochondrial population, a fundamental aspect allowed by the mitochondrial quality control system, in which mitochondrial dynamics, mitochondrial biogenesis, and mitochondrial autophagy (named mitophagy) are the main events [106]. Fusion and fission help to keep mitochondrial structure integrity and are necessary steps to improve metabolism and facilitate cooperation and communication between mitochondria [107]. At the same time, erroneous fission and fusion events may provoke the formation of a nonfunctional mitochondria population, which can induce the production of damaged elements (such as mitochondrial reactive oxygen species) that are harmful for the entire mitochondria population and for the cell. Injured mitochondria can be removed by the degradative process, mitophagy, which recognizes and sequesters the damaged organelle into autophagy vesicles, which are then delivered to the lysosome for degradation [108]. Finally, preserving the adequate mitochondrial number of the cell may intervene in mitochondrial biogenesis that is responsible for generating new mitochondrial offspring [109]. Decline and/or sustained activation of these molecular mechanisms deputed to control the mitochondrial amount and quality has been associated with several human diseases, in particular, mitochondrial diseases [110], genetic disorders [111], neurodegeneration [112,113], cancer [114], and cardiovascular disorders [46].

For instance, immunostained calcified human AVs revealed high levels of the mitochondrial fission initiator protein dynamin-related protein 1 (DRP1). Inhibition of DRP1 via RNA interference promotes a reduction in the osteogenic differentiation process and inhibits oxidative stress [115]. A more recent study highlighted the overexpression of protein tyrosine phosphatase 1B (PTP1B) in CAVD [116]. PTP1B is a negative regulator of the leptin and insulin signaling pathways, which are involved in the regulation of mitochondrial dynamics and biogenesis [117,118]. The authors demonstrate the decreased osteogenic differentiation of interstitial valvular cells by the pharmacological inhibition of PTPB1. This effect is accompanied by an upregulation of mitochondrial biogenesis, which has been observed to be downregulated during the progression of valvular calcification [116]. Morciano et al. revealed the presence of aged mitochondria together with reduced PGC-1α expression, a key mitochondrial biogenesis protein, in interstitial cells isolated from human patients [46]. Impairment of mitochondrial biogenesis in CAVS is associated with increased cell death and the presence of aged mitochondria, despite an increase in mitophagy and autophagy fluxes, suggesting an insufficient turnover of mitochondria in CAVD samples. Moreover, the authors revealed for the first time the presence of other mitochondrial impairments in CAVD patients, such as calcium dysregulation, reduced respiratory capacity, and lack of ATP production [46].

The gene expression profile of AV tissue identified several novel genes associated with mitochondrial functions variations that are involved in the pathogenesis of CAVD, such as an increase in reactive oxygen species (ROS) production and reduced mitochondrial membrane potential, metabolic imbalance, and mitochondrial fragmentation [119]. Moreover, the authors suggest that integrated miRNA/mRNA analyses might be used as diagnostic biomarkers for CAVD [119]. Interestingly, they later identified matrix metalloproteinase 9 expression (MMP9), a mitochondrial-related gene, as being extremely high in AS samples, which would be a useful biomarker for aortic stenosis [120].

Inside the cell, Calcium (Ca^2+^) is one of the major second messengers and regulates a plethora of biological processes, such as metabolism, antioxidant defense, apoptosis, muscle contraction, neurotransmitter release, and also mitochondrial functioning [121]. Therefore, it is not surprising that dysregulations of Ca^2+^ dynamics are involved in different pathologies [122]. In the context of aortic calcification, evidence that connects aortic stenosis and Ca^2+^ dysregulation goes back to a genome-wide association study [123], where calcified valves reveal the upregulation of mRNA levels of RUNX2 concomitant with an increased calcium voltage-gated channel subunit alpha 1 C (*CACNA1C*) gene [123], which encodes the CaV1.2 L-type voltage-gated Ca^2+^ channel. Later, these data were confirmed by another study, which demonstrated a higher Ca^2+^ influx in CAVD patients through the CaV1.2 channel [124].

VEGF is a target gene of HIF-1α that regulates several cellular processes including proliferation, cell survival, differentiation, and migration [125]. Xu et al. have demonstrated the impact of VEGF in sustaining the mitochondrial fission and fusion balance, mitigating the mitochondrial apoptotic pathway; thereby, VEGF could play a critical role in repairing the AS transition from compensatory cardiac hypertrophy to HF in mouse animal models [125].

It is important to note that patients with AS exhibit a metabolic shift from fatty acid to glucose metabolism, which is characterized by a decreased expression of fatty acid translocase (FAT/CD36) protein, together with a downregulation of other fatty acid transporters, such as plasma membrane and heart-type cytosolic fatty acid binding proteins (FABPpm and H-FABP), β-oxidation, Krebs cycle, and oxidative phosphorylation proteins. On the contrary, the same cardiac biopsy exhibits an increased expression of glucose transporter 1 and 4 (Glut 1, 4). This study suggests a downregulation of fatty acid oxidation proportionally to a more severe outcome in patients with aortic stenosis [126]. Nonetheless, whether this metabolic shift in CAVS patients results from the involvement of mitochondrial ROS (mtROS) in instigating AV calcification or whether another mediator contributes to this process remains unknown.

Notably, such a metabolic shift to glycolysis is well demonstrated in various CVDs orchestrated by HIF-1α; however, its effect on the metabolism adaptation in CAVD is yet to be explored. HIF-1α regulates multiple genes influencing mitochondrial activity and is crucial for the metabolic shift, such as LDH-A and phosphoglycerate kinase-1 (PGK1) [127], thus elevating anaerobic glycolysis by enhancing the generation of glycolysis enzymes, increasing glucose transporters expression (e.g., Glut 1, 4), and inhibiting mitochondrial energy metabolism [128,129], while impeding fatty acid oxidation [7,130]. Several studies have supported the notion that HIF-1α plays a significant role in oxidative stress, as its expression level is intrinsically associated with mtROS in response to O_2_ deprivation [7,131].

Furthermore, it has been reported that human calcified valves are enriched in oxidative stress, worsening the progression of calcification [132]. The increased oxidative stress levels seem to be inversely proportional to antioxidant enzyme expression and functionality, as well as the uncoupling nitric oxide synthase (NOS) activation [133,134]. Together these data illustrate the crucial role of mitochondria in the pathophysiology of CAVD.

### 3.4. Relationship between Inflammation and Mitochondria in CAVD

As we reviewed in the previous paragraphs, the inflammatory response has been the only known mediator of CAVD in the past [135,136], and its connection with the onset and progression of the disease that resulted in either surgical valve replacement or transcatheter AV implantation has been extensively reported [26,137]. Indeed, a significant amount of clinical data, extrapolated from patients affected by AS, showed with different techniques, high levels of inflammation. In the 1990s, aortic valvular lesions from this cohort of patients were carefully analyzed through popular biochemical approaches; the native tissue appeared as being characterized by thickening, by a large amount of lipids deposition, and by the presence of mineralization and calcium accumulation into the leaflet (Figure 3). Concurrently, a high grade of inflammatory infiltrate with foam cell macrophages was present in the lesions [138]. From plasma samples of AS patients, IL-1β and IL-18 were the main circulating cytokines to be overexpressed [46]; however, IL-6 and TNF have also been reported to play a role in osteogenic differentiation with a significant pool of M1 polarized macrophages [139]. In 2012, Dweck MR and co-workers used for the first time the positron emission tomography (PET) and two common PET tracers, 18F-Flurodeoxyglucose (18F-FDG) and 18F-Sodium fluoride (18F-NaF), which were able to target calcification and inflammation into the same patient [140]. They found the inflammatory response increased by 91% in AS patients compared to healthy subjects [140].

Today, inflammation is still considered to be the main active player in the phases that precede calcification, but it is deeply nestled among other major contributors such as mitochondrial dysfunctions for which cause–effect relationships are difficult to address [26,46,141,142].

The link between mitochondria and inflammation is very close. A seminal paper by Zhong Z. and co-workers showed how mitochondria directly orchestrate signals from TLR to produce oxidized mtDNA fragments, needed to activate NLRP3 inflammasomes once they have moved into the cytosol [143]. Basically, according to this evidence, mitochondria constitute a reservoir of many signaling substances. A pool of these, once released into the cytosol, becomes harmful and can trigger mitochondria-mediated inflammation through multiple receptor-dependent responses [144]. These are the so-called mitochondrial danger-associated molecular patterns (mtDAMPS) and include, for example, ATP, mtDNA, and mtROS.

NLRP3 is a multiprotein platform that is activated by mtDAMPS; it relocalizes to mitochondria and associated membranes and plays a crucial role in CAVD through the production of the mature form of IL-1β [145]. This cytokine is consistently upregulated in CAVD patients, is highly expressed in situ in calcified areas, and further sustains inflammation, stimulating IL-6 and IL-8 production and activating the NF-kB pathway [146]. It is documented that IL-1β activates HIF-1α to stimulate a metabolic shift from oxidative phosphorylation (OXPHOS) to glycolysis, which is crucial in adaptive immunity [147]. The exacerbation of the inflammatory phenotype is also given by the enhancement of the MPP1 function with drastic changes in the extracellular matrix composition [148]. Accordingly, the antagonization of the IL-1 receptor by the IL-1 receptor antagonist (Ra) has been shown to block the proinflammatory pathway [149], conferring on the receptor properties targeted for cardioprotection. Moreover, its genetic depletion in animal models of CAVD has definitively shown its crucial contribution to the calcification process of the AV [149].

Additionally, to the aforementioned studies in the previous sections, HIF-1α upregulates mtROS levels, stimulates the NF-kB transcription factor, and activates inflammasome genes expression, including NOD-, LRR-, and pyrin domain-containing protein (NLRC)4, NLRP3, and the IL-1β genes. Thereby, leading to mitochondrial oxidative stress, affecting in return mitochondrial membrane permeability, lipid peroxidation, and mtDNA, as a consequence of mitochondrial abnormalities [7].

Mitochondria are also a place where ROS are produced in great amounts, mainly as end-products from OXPHOS [150]. ROS are usually culprits of mitochondrial damage in cardiovascular diseases, especially when an overproduction of free radicals or impairment of the ROS-scavenging enzyme system occurs [151,152,153,154]; moreover, ROS further sustain inflammation [155]. In the last few years, evidence involving mtROS in CAVD has become manifold. First studies carried out on isolated ex vivo cultures from human valve samples demonstrated how the presence of lipoprotein a (Lp(a)) selectively triggered acute superoxide production from mitochondria, [156] and through it, Lp(a) chronically lead to cell calcification. Lp(a) induces significant calcium deposition in vitro, and patients with CAVD have higher plasma levels of Lp(a) [157,158]. mtROS reached high levels in a few hours after Lp(a) exposure, but the calcification followed in several days. This evidence ascribes a role for mtROS generation in the acute stages of the disease, but it is still in doubt if they can be targeted to revert the pathological phenotype once the CAVD condition is established.

In addition to Lp(a), another apolipoprotein named ApoC-III was found to be involved in functions far from their primary role in metabolism [141]. Similarly, ApoC-III was detected in large amounts around the calcific regions of human AV leaflets and was directly related to the calcification grade [141]. The process was further monitored in vitro through the chronic treatment of normal noncalcified interstitial cells with ApoC-III, which led to calcification in a few days. The pathological process was accompanied by an increase in several mitochondrial proteins involved in oxidative stress and in the inflammatory cascade with the upregulation of IL-6 and BMP-2 expression [141]. Among the signs of mitochondrial stress during calcification, the expression of superoxide dismutase 2 (SOD2), HSP60, HtrA Serine Peptidase 2 (HTRA2), and Caseinolytic Mitochondrial Matrix Peptidase Proteolytic Subunit (CLPP) are the most affected proteins. They usually participate in apoptotic and mitophagic pathways [159]. Inflammation in CAVD also acts via TNF-α. There was an acute increase in the number of cytokines at 30 min and chronically, at 21 days, associated with a significant induction of mtROS [160]. These results were only observational without any correlation to the calcific phenotype. Besides the involvement of HSP60, the downregulated expression of HSP90 has also been revealed in a proteomic analysis of CAVD, leading to ROS generation and endothelial alterations [161].

The organized interplay between mtROS and inflammation also participates in the calcification process by regulating the osteogenic differentiation of vascular smooth muscle cells (VSMC) through the fine-tuning of RUNX2 expression [162]. Furthermore, mtROS are generated by mitochondrial carbonic anhydrases, enzymes that result in upregulation in calcified aortic tissues [163]. The inhibition of their function alleviates both inflammation and the calcification process to which VSMCs are subjected [163].

Besides this partially direct evidence of the contribution of mitochondria to inflammation through mtROS release and linked to mitochondrial stress, a study by Smyrnias, in 2019, described the presence of a dysfunctional mitochondrial unfolded protein response (mtUPR) in patients affected by AS. The role of mtUPR in heart and cardiac diseases is unclear, but this work showed how patients having a higher activation of mtUPR also have a lower rate of myocyte death and fibrosis/calcification levels [164].

## 4. Novel Insights on the Therapeutic Strategies of CAVS

Despite the increasing amount of evidence to better comprehend the factors underlying CAVS progression. Nowadays, this disorder is a major growing challenge, and it remains unclear in terms of molecular and cellular mechanisms, and presently no therapeutic strategies successfully exist to prevent or treat CAVS. Only surgical intervention or transcatheter valve replacement is an efficient treatment option for CAVS. Nevertheless, it is well recognized that these types of therapeutic options are highly complicated, in which the surgical implantation of an artificial valve requires lifelong antithrombotic therapy, and the bioprosthetic valves are prone to deterioration over time, requiring patients to have a further operation, as well as a higher risk of thromboembolism due to the reduced flow and pressure [165,166]. To the best of our knowledge, at present, no therapeutic strategies successfully exist yet to prevent or treat CAVS; therefore, the development of a potent medical therapy for CAVS becomes a major urgent necessity to reverse the development of the disease and ameliorate the clinical outcomes. To attain this noble purpose, it is highly necessary to deeply focus and target the molecular/cellular mechanisms for a better understanding of the etiopathology. In Table 1, we report an updated list of the most promising therapeutic options for CAVS.

In recent years, so much has been learned about novel promising pharmacotherapy for CVDs. From the pool of molecules, one favorable molecule is HIF-1α, which has been shown to act as a cardio-protectant in different aspects of CVDs. However, very little has been explored regarding its role in CAVS. A few studies showing the involvement of HIF-1α in CAVS are addressed in the earlier sections of this review but none have demonstrated HIF-1α’s role in the staging of calcification. It has been very recently discovered that HIF-1α expression can be blocked by PX-478 and miR-483 mimics, which are novel stretch-sensitive and flow-sensitive disease molecules, respectively. Thus, they convey a novel potent target to develop an innovative therapeutic strategy to alleviate CAVD [58,167]. Further studies are required to study the different members of the HIF family in CAVS.

Dimethylloxetane, an inhibitor of HIF-1α, downregulates osteogenic transcription factors including RUNX2 and BMP-2, thereby reducing vascular calcification in ovariectomized rats. In addition, estrogen drug therapy alleviates HIF-1α expression and vascular calcification [168]. Indeed, studies have also documented the role of estrogen as a negative modulator of several mechanisms, such as inhibiting NFκB signaling and the receptor–activator of nuclear factor κB ligand (RANKL) [71,130], blocking p53 activity [169], inhibiting the activity of NADPH oxidase, and causing inflammation [24,153]. Moreover, regarding its role mentioned previously, as a negative modulator, estrogen can direct the activation of antioxidant enzymes in mitochondria, in the lysosome, and in the cytosol to enhance NOX expression and contribute to the protective impact in calcification [170,171]. Another inhibitor of HIF-1α that still needs to be better investigated in cardiovascular calcification is the HIF-1α mRNA antagonist EZN-2968, [172]. Studies are necessary to unveil whether it could be a therapeutic alternative to treat vascular calcifications in the future.

Hypoxia has been reported to contribute to the early development of the disorder and stimulate HAVIC proliferation. Hypoxia is a canonical stimulator of the angiogenesis process through HIF-1 transcription activity. A novel correlation between hypoxia and pro-angiogenic conditions has been recently demonstrated in CAVS [65]. Furthermore, hypoxia modulates angiogenesis in CAVS through the activation of VEGF receptor sFlt1 expression, which is an anti-angiogenic component where its released mechanism action is still unknown [65]. Inhibiting the angiogenesis process has demonstrated a potential target to treat various disorders, such as cancer [173,174], and CVD including myocardial infarction [175]. In a recent study, S1P is identified to halt the formation of neo-vessels by targeting sFlt1 in CAVS [65], suggesting further future therapeutic approaches to stimulate the S1P receptors in CAVS to limit the progression of the disorder. Treatments used for vascular problems have been suggested to ameliorate CAVS outcomes; however, it is well-known that the molecular mechanisms of CAVS are different from vascular disorders [176].

Angiotensin-II (Ang-II) plays a crucial role in CAVS progression. It upregulates IL-6 generation to stimulate cardiac fibrosis and to stimulate the osteogenic differentiation of valve interstitial cells [177,178]. Inhibitors of Angiotensin-converting enzyme (ACE-I) or Ang-II receptors have been widely used for hypertension treatment and have been explored in CAVS progression and the multiple severities of CAVS [179,180]. ACE-I therapy is correlated with reduced CAVS and a slowdown in mild CAVS development [181,182]. Indeed, ACE-I therapy has shown amelioration in the hemodynamic and left ventricular (LV) hypertrophy in patients with moderate and severe CAVS in randomized clinical trials, for example, the RIAS [183,184]. In the same regard, potent blockers of Ang-II such as fimasartan and losartan have shown a beneficial impact in severe CAVS patients with hypertension (clinicaltrials.gov/NCT03666351). Moreover, the targeting of renin–angiotensin–angiotensinogen system (RAAS) inhibitor therapy may impede CAVS progression and reduce the mortality risk in CAVS patients [185]. Thus, large-scale trials are needed to dissect this further therapy in delaying the development of CAVS.

KPT-330 is a potent selective exportin-1 (XPO1) inhibitor, widely involved in cancer disorders, which regulates the expression of cyclin D1 [186]. KPT-330 has shown an effective impact in vitro to mitigate calcific nodule formation protecting against CAVS through a novel CCAAT/enhancing-binding protein (C/EBPβ) signaling pathway, suggesting KPT-330 as an alternative pharmaco-based therapy to treat CAVS [187].

Melatonin (N-acetyl-5-methoxytryptamine) is a pleiotropic molecule with cardioprotective effects, maintains mitochondrial function via its antioxidant activities, and impedes the opening of the permeability transition pore complex (PTPC) [188]. Notably, PTPC is a mitochondrial supramolecular entity [189] that initiates mitochondrial permeability transition, an event characterized by a sudden and irreversible augmentation of the permeability of mitochondrial membranes, which causes the dissipation of the mitochondrial transmembrane potential, osmotic breakdown of the organelle, and, finally, cell death [190]. In addition, melatonin targets and shields the mitochondrial fission protein DRP1 in diabetic hearts [191], and blocks DRP1 in HUVEC through the stimulation of the AMPK/Sarco-Endoplasmic Reticulum Calcium ATPase (SERCA) signaling pathway [192]. Besides its effective roles in mitochondria, melatonin plays an important role as an anti-inflammatory effect [193]. Recently researchers have shown much interest in studying the cardioprotective capabilities of melatonin in CAVS. Melatonin has been shown to improve CAV in valve interstitial cells via RNA CircRIC3/miR-204-5p/DPP4 signaling [194], and it targets melatonin receptor (MT1)/NF-κB/RUNX2 signaling to reduce the calcification in valve interstitial-mediated osteogenic stimulation [195]. Another significant study shows the crucial protective impact of melatonin in inhibiting calcification through the AMPK/DRP1 pathway in vascular smooth muscle cells (VSMCs) [196].

The enzyme carnitine O-octanoyltransferase (CROT) is responsible for the transport of medium and long-chain acyl-CoA molecules out of the peroxisome to the cytosol and mitochondria. antiCROT has been discovered as a novel recent candidate that contributes to valve fibrocalcification. The inhibition of CROT corrects mitochondrial abnormalities in VICs by blocking mitochondrial fragmentation, restoring mitochondrial proteome changes in the osteogenic environment, and promoting fatty acid metabolism [197].

Recently, microarray technologies have been used to identify newly discovered functional genes that can also be adopted for ultimate diagnosis and prognostics in various diseases [198]. Among these, microarray profiling identified the proprotein convertase subtilisin/kexin type 9 (PCSK9), an enzyme mainly secreted in the liver that targets the LDL receptor for further lysosomal destruction, which is involved in the inflammatory gene’s expression, cell cycle, and stress condition responses [199]. Additional studies demonstrated the impact of PCSK9 in promoting the inflammatory response, which is the main contributor to valve calcification, through the stimulation of T cells infiltrating calcified AVs and promoting inflammation; besides its role in activating the T cells, PCSK9 can also stimulate directly macrophages to secrete proinflammatory cytokines including, IL-6, IL-1β, and TNFα [200,201]. Ongoing clinical trials are presently testing whether blocking PCSK9 may mitigate the micro/macro-calcification of AVs using computed tomography and 18F-NaF positron-emission tomography/computed tomography, respectively, in CAVS patients (clinicaltrials.gov/NCT03051360). Indeed, PCSK9 inhibitors, such as evolocumab, in the explorative analysis of the “FOURIER” trial (Further Cardiovascular Outcomes Research with PCSK9 Inhibition in Subjects with Elevated Risk) [202], demonstrated positive outcomes in reducing CAVS events after the first year of the therapy [143].

Other crucial genes that have been recently discovered in CAVS pathogenesis through advanced gene chip technology are the Secretogranin II (SCG2) and C-C motif chemokine ligand 19 (CCL19), which are involved in inflammatory responses; however, their exact correlation and mechanism with the CAVS are still to be elucidated. As previously reported by Fang and colleagues, SCG2 alters angiogenesis and tumor growth by HIF-1α destruction [203] and is able to stimulate wound healing in response to fasting [204]. These findings suggest a further impact of SCG2 in tissue remodeling and provide further insights for future research into immunotherapy strategies for CAVS progression.

On the other hand, MSI-1436, a pharmacological inhibitor of PTPB prevents fibrocalcification of the AV and protects the mitochondrial biogenesis and performances through the optic atrophy 1 (OPA1) regulation [116].

Recent studies by Liu et al. demonstrate that MMP-9i, an inhibitor of MMP9, reveals its significant effectiveness in attenuating mitochondrial impairments and repressing oxidative stress, and thereby VIC calcification in CAVS patients [120].

Rapamycin is a potent inducer of autophagy, and it has been shown by Morciano and colleagues to revert calcification by attenuating cell death and restoring calcium dysregulation in CAVS patients [46]. It is, however, required to be studied in animal models in combination with an anti-inflammatory.

**Table 1 ijms-24-11105-t001:** A representative table on the therapeutic strategies of CAVS.

Therapeutic Target	Treatment	Mechanism of Action and Effect	References
HIF-1α inhibitors	Dimethylloxetane	Downregulates osteogenic transcription factors (RUNX2 and BMP-2), reduces vascular calcification	[168,205]
	PX-478	Downregulates porcine AV (PAV) calcification, and significantly blocks the deposition of calcium, thereby alleviating CAVD	[58,167]
	EZN-2968	Possible to mitigate the calcification	[172]
Estrogen drug therapy	-	Alleviates HIF-1α expression and vascular calcification, inhibits NFκB signaling and RNKL, blocks p53 activity and NADPH oxidase, as well as inflammation. Activates antioxidant enzymes including in mitochondria and lysosomes, as well as the cytosol	[168,169,170,206,207,208,209]
Angiogenesis repressor	S1P	Halts neo-vessels formation via sFlt1 activation	[65]
RAAS inhibitors (ACE-I therapy Ang-II blockers)	Fimasartan and losartan	Reduces CAV and a slowdown in mild CAVS development. Improves hemodynamic and left ventricular (LV) hypertrophy in patients with moderate and severe CAVS	[179,180,181,182], https://clinicaltrials.gov/show/nct01589380, https://clinicaltrials.gov/ct2/show/NCT03666351 (accessed on 30 March 2023)
Exportin-1 (XPO1) inhibitor	KPT-330	Mitigates calcific nodule formation via CCAAT/enhancing-binding protein (C/EBPβ) signaling pathway, protecting against CAVS	[187]
Melatonine (N-acetyl-5-methoxytryptamine)	Melatonine	Improves CAV in VICs via RNA CircRIC3/miR-204-5p/DPP4 signaling and targets MT1/NF-κB/RUNX2 signaling, reducing VIC calcification-mediated osteogenic stimulation. Inhibits calcification via the AMPK/DRP1 pathway in VSMCs	[194,195,196]
CROT inhibitor	-	Blocks mitochondrial fragmentation, restores mitochondrial proteome changes in the osteogenic environment, and promotes fatty acid metabolism, and reduces fibrocalcification	[197]
PCSK9 inhibitors	Evolocumab	Reduces CAVS events	[143,202]
PTPB inhibitor	MSI-1436	Prevents AV fibrocalcification, protects mitochondrial biogenesis, and performances via OPA1	[116]
MMP9 inhibitor	MMP-9i	Attenuates mitochondrial impairments, represses oxidative stress, and VIC calcification	[120]
Autophagy activator	Rapamycin	Reverts calcification through cell death downregulation and restores calcium dysregulation in CAVS patients	[46]

## 5. Conclusions and Future Perspective

CAVS is a complex and progressive heart valvular disease. Extensive research has led to the identification of several risk factors underlying CAVS progression. The early insidious nature and etiology of CAVS have made it challenging to explore and unravel the cellular and molecular mechanisms underlying its complex etiology. This has left us with only surgical and transcatheter valve replacement as the only possible option to treat CAVS, which is very likely a costly procedure that is expected to double by 2050 [6]. At present, no therapeutic approaches have been successfully developed yet to prevent or treat the disease. Therefore, the development of a potent therapy for CAVS has become a major urgent and critical necessity. To attain this noble purpose, it is highly essential for scientists to dive deep into understanding the molecular and cellular mechanisms of CAVS initiation and progression.

Inflammatory pathways are one of the major molecular mediators in both the initiation and progression of CAVS. Recent insights demonstrate that impairments in the functioning of mitochondria are involved in the pathogenesis of this disorder through both the processes of oxidative stress and inflammation. Furthermore, discoveries have identified the hypoxia signaling pathway as a critical contributor in the early and evolution stages of CAVS. However, extended investigations are required to better dissect the hypoxia-related pathways in CAVS and their interconnections with mitochondria and inflammation. Another important point to inspect is how members of the HIF1 family can regulate the hypoxic levels to modulate the different processes of autophagy and mitophagy in CAVS, taking into account of the important role of these pathways in controlling mitochondrial homeostasis and inflammation.

Understanding how elements related to the HIF1 family contribute to CAVS disease could help to solve another great issue in the study of CAVS, which is the lack of established elements for an early diagnosis and prognosis. Furthermore, assessing whether the markers of hypoxia can influence the calcification score in calcific AV areas of patients by applying artificial intelligence would be an innovative approach to predict the severity and extent of calcification, and to be used for further prognosis.

The identification of the close relationship between hypoxia signaling, mitochondrial dysfunction, and inflammation in CAVS could be also essential to develop new therapeutic approaches that target factors of these signaling pathways at the early phases of the disease rather than at the end stage where patients exhibit severe aortic stenosis.

## Figures and Tables

**Figure 1 ijms-24-11105-f001:**
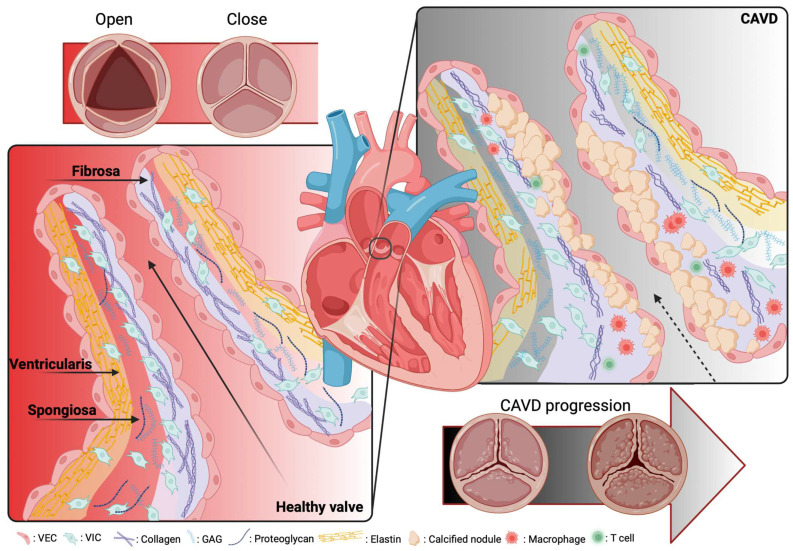
Schematic representation of a native healthy and calcific aortic valve. The healthy aortic valve contains 3 layers of extracellular matrix termed fibrosa that consist of collagen, on the aorta side, a central layer named spongiosa contains proteoglycan and glycosaminoglycans, and a layer named ventricularis consists of elastin fiber, on the left ventricle side. The main cells in the aortic valve cusp core are the endothelial and the interstitial cells (the left) Therefore, in the pathological case, the valve cusp becomes thick and calcified on the surface of the fibrosa, with further fragmentation in the elastin fibers and the enhanced collagens (the right).

**Figure 2 ijms-24-11105-f002:**
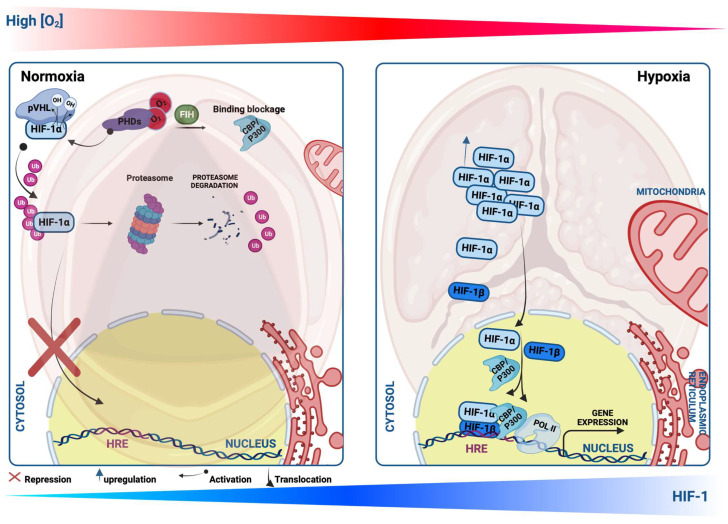
Schematic illustration showing Hypoxia-inducible factor-1 α (HIF-1α) protein regulation during normoxia and hypoxia. Under the normoxic condition, HIF-1α protein is hydroxylated by propyl-hydroxylases (PHDs) and factor-inhibiting HIF (FIH), which facilitate the binding of HIF-1α with the von Hippel–Lindau protein (pVHL), leading to its ubiquitination, and thus proteasomal degradation. Upon the PHD inhibition or hypoxic condition, HIF-1α translocates to the nucleus, where it heterodimerizes with HIF-1β and binds to a core putative sequences of target genes termed hypoxia response element (HRE) and stimulates their transcriptional activity.

**Figure 3 ijms-24-11105-f003:**
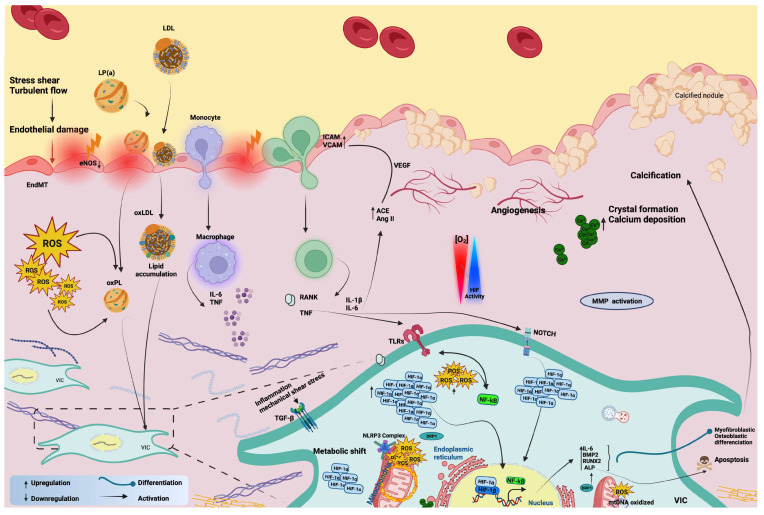
A diagram representing the contribution of inflammation, mitochondrial dysfunction, and hypoxia signaling in the pathophysiology of the calcific aortic valve.

## Data Availability

Not applicable.

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
