# Peer review of "The Complex Relationship between Hypoxia Signaling, Mitochondrial Dysfunction and Inflammation in Calcific Aortic Valve Disease: Insights from the Molecular Mechanisms to Therapeutic Approaches"

_ijms, 2023, doi:10.3390/ijms241311105_

Round 1
Reviewer 1 Report
I have read with great attention and interest the review paper entitled "The Complex Relationship between Hypoxia Signaling, Mitochondrial Dysfunction and Inflammation in Calcific Aortic Valve Disease: Insights from the Molecular Mechanisms to Therapeutic Approaches"
The authors have carried out an elegant, accurate and very well organized narrative review.
I have only one minor suggestion for the Authors: the paper is very long and verbose, it should be shortened, especially introduction and discussion.
The abstract, on the other hand, is overall unclear: I would suggest the Authors to rewrite it, briefly explain the reason for this review paper, its findings, and possible future developments
Author Response
Reviewer 1
Comments and Suggestions for Authors
I have read with great attention and interest the review paper entitled "The Complex Relationship between Hypoxia Signaling, Mitochondrial Dysfunction and Inflammation in Calcific Aortic Valve Disease: Insights from the Molecular Mechanisms to Therapeutic Approaches"
|
The authors have carried out an elegant, accurate and very well organized narrative review. |
|
- We thank the reviewer for the kind and the valuable comment. |
|
I have only one minor suggestion for the Authors: the paper is very long and verbose, it should be shortened, especially introduction and discussion. |
|
- Thank you, we have addressed this point as requested. |
|
The abstract, on the other hand, is overall unclear: I would suggest the Authors to rewrite it, briefly explain the reason for this review paper, its findings, and possible future developments |
|
- We thank the reviewer for the suggestion, we have modified the abstract accordingly. |
Reviewer 2 Report
With great pleasure I would address several comments about the paper The Complex Relationship between Hypoxia Signaling, Mitochondrial Dysfunction and Inflammation in Calcific Aortic Valve Disease: Insights from the Molecular Mechanisms to Therapeutic Approaches.
The main question of the paper is the impact of tissue hypoxia on degenerative aortic stenosis formation. The topic is original and relevant. The paper is the most essential and updated review on the topic to the date.
Conclusions are consistent with the review body. Notably, authors include the Table 1 with current data on available treatment approaches. References are updated , but have to be formatted in the text body (numbers). Central illustration is good. References in the Table 1 have to be corrected I highly recommend this paper for publication.
English is acceptable.
Author Response
Reviewer 2
Comments and Suggestions for Authors
|
With great pleasure I would address several comments about the paper The Complex Relationship between Hypoxia Signaling, Mitochondrial Dysfunction and Inflammation in Calcific Aortic Valve Disease: Insights from the Molecular Mechanisms to Therapeutic Approaches. |
The main question of the paper is the impact of tissue hypoxia on degenerative aortic stenosis formation. The topic is original and relevant. The paper is the most essential and updated review on the topic to the date.
|
Conclusions are consistent with the review body. Notably, authors include the Table 1 with current data on available treatment approaches. References are updated, but have to be formatted in the text body (numbers). Central illustration is good. References in the Table 1 have to be corrected I highly recommend this paper for publication.
|
|
- We thank the reviewer for the observation, we have modified the references accordingly. |
Reviewer 3 Report
The present review aims to cover the most recent understanding of the impact of the complex interplay effect of hypoxia signaling pathways, mitochondrial dysfunction, and inflammation on a key condition of cardiovascular disease, which is "calcific aortic valve stenosis”.
1. The article is novel and interesting and addresses an hot topic in the current literature.
2. The Introduction is well written and with good references.
3. Although this is a general review a short paragraph at the end of introduction can be placed in order to give some information on methodology used to search articles (i.e., inclusion and exclusion criteria, time frame, databases used, type of articles included, etc).
4. All the sections of the article are well written, but the last (5. Conclusion and Future Perspective) that is too short in the current format and needs to be expanded with more critical insight.
5. English should be improved for style.
English should be improved for style.
Author Response
Reviewer 3
Comments and Suggestions for Authors
The present review aims to cover the most recent understanding of the impact of the complex interplay effect of hypoxia signaling pathways, mitochondrial dysfunction, and inflammation on a key condition of cardiovascular disease, which is "calcific aortic valve stenosis”.
Comments:
|
1. The article is novel and interesting and addresses an hot topic in the current literature. |
|
- We thank the reviewer for the valuable comment.
|
|
2. The Introduction is well written and with good references. |
|
- We thank you for your honest comment that we appreciate. |
|
3. Although this is a general review a short paragraph at the end of introduction can be placed in order to give some information on methodology used to search articles (i.e., inclusion and exclusion criteria, time frame, databases used, type of articles included, etc). |
|
- We thank the reviewer for the recommendation, we have included these points accordingly. |
|
4. All the sections of the article are well written, but the last (5. Conclusion and Future Perspective) that is too short in the current format and needs to be expanded with more critical insight. |
|
- Thank you for pointing this out, we revised this part as recommended. |
|
5. English should be improved for style. |
|
- Thank you for pointing this out, we revised this part as recommended. |
Round 2
Reviewer 3 Report
amended manuscript is acceptable